# Modeling Architectural Heritage in Wikidata: A Case Study of Early Modern European Hospitals

**Authors:**

*Frieder Leipold, Email:* [F.Leipold@kunstgeschichte.uni-muenchen.de](mailto:F.Leipold@kunstgeschichte.uni-muenchen.de)

*Maximilian Kristen, Email:* [max@kristenonline.de](mailto:max@kristenonline.de)

*Lily Marie Baumeister, Email:* [L.Baumeister@campus.lmu.de](mailto:L.Baumeister@campus.lmu.de)

*Isabella Limmer, Email:* [isabella.Limmer@campus.lmu.de](mailto:isabella.Limmer@campus.lmu.de)

*Chiara Franceschini, Email:* [Chiara.Franceschini@kunstgeschichte.uni-muenchen.de](mailto:Chiara.Franceschini@kunstgeschichte.uni-muenchen.de)

*Common address:* Institut für Kunstgeschichte, Zentnerstraße 31, 80798 München

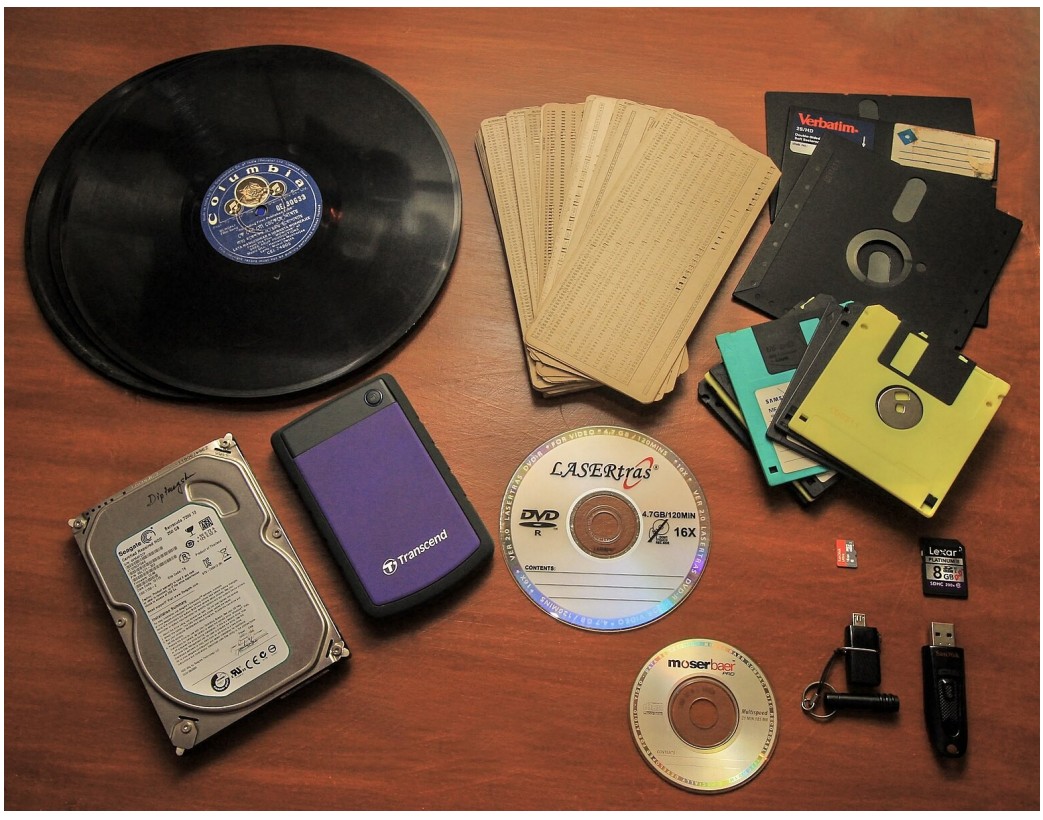

*Figure 1: Evolution of Storage Media: Records, Punched Cards, Floppies, Hard Disk, DVD/CD, USB Flash Drives, Memory Cards; Diptangshudatta on [WikimediaCommons](), licensed under CC-BY-SA-4.0*

# Introduction

The storage devices depicted in this image (fig. 1) illustrate the **evolution of data storage** from the 20th century to the present day. Starting with analogue vinyl records, moving through punched cards for early computing machines, to floppy disks, hard drives, DVDs/CDs, USB flash drives, and memory cards. Now, ask yourself: which of these storage devices could I still read in my office today? Digital technologies enabled data storage at an unprecedented scale. However, with this digital media, data loss is always a risk. Every time we switch to a new medium, data that isn't migrated to the new environment is lost.

This simple example illustrates a larger and increasingly urgent issue in the digital humanities: the **fragility and transience of research data**. As digital methods have become central to academic research, the long-term sustainability of the data we produce has emerged as a major challenge. Software updates, changing file formats, and the obsolescence of storage media all threaten the accessibility of valuable research data. Compounding the problem, many research databases are developed as part of time-limited projects. When funding ends, so does the hosting—and with it, access to the data itself.

The ARCHIATER project, funded by the European Research Council, tries to establish a compelling response to this crisis. It treats Wikidata not as a passive archive but as a living, collaborative, and sustainable **Virtual Research Environment (VRE)**. In this paper, we present how ARCHIATER integrates Wikidata as a central component of its research methodology to document and analyse the visual and architectural culture of premodern hospitals in Europe. Emphasizing the benefits of both qualitative and quantitative approaches, we demonstrate how Wikidata supports long-term scholarly collaboration, transparency, and innovation.

Drawing on lessons learned from earlier digital projects—such as the 3D reconstruction of Schloss Weikersheim and the DeJongeWiki on Schloss Arenberg—ARCHIATER joins the broader WikiFAIR initiative. This initiative advocates for academic data to be hosted on Wikimedia platforms, ensuring compliance with the **FAIR principles: Findable, Accessible, Interoperable, and Reusable**. We argue that Wikidata provides a robust infrastructure for the humanities and can serve as a model for future digital research projects.

# The ARCHIATER Project: Goals and Framework

ARCHIATER stands for "Heritage of Disease: The Art and Architectures of Early Modern Hospitals in European Cities."[1] The five-year project, based at Ludwig-Maximilians-Universität München and funded through an ERC Grant, explores the architectural, artistic, and religious dimensions of **hospital buildings from the 14th to 18th centuries**. These institutions were far more than centres of medical care. They were hubs of public life and visual culture, richly furnished with religious symbols, artworks, and civic emblems.

Focusing on the countries of pre-modern Europe, ARCHIATER explores how **hospitals reflected and shaped their urban environments**. While differing in regional contexts, these institutions share common traits as complex cultural nodes. By adopting a multidisciplinary approach that includes art history, architecture, religious studies, and digital humanities, the project seeks to illuminate the pivotal role hospitals played in shaping early modern societies.

So, we are dealing with a research subject that is vast both temporally and geographically, a subject that developed very complex structures. Additionally, ARCHIATER is a collaboration between scholars at the universities of Munich in Germany and Lucca in Italy. This raised the question of how the **ongoing research progress** could be documented and how the emerging data connections could be stored in the medium term.

Although there are highly advanced tools for data processing, even in a scientific context, these offerings are often associated with paid licenses. If you no longer pay, you lose access. Furthermore, you are dependent on the financial stability of the respective service provider.

# Data Fragility in Historical Research

Despite the growing sophistication of digital tools, many research datasets have a short shelf life. Projects often develop bespoke databases or digital platforms tailored to their research questions. These platforms may offer **short-term functionality**, but they are rarely maintained after funding ends. Without continued investment, they fall victim to outdated software, broken links, and **incompatible data structures**.

These problems are compounded by data silos. Custom platforms are often poorly integrated with one another. Their data cannot be easily shared, reused,

---

[1] ARCHIATER, Heritage of Disease: The Art and Architectures of Early Modern Hospitals in European Cities, https://cordis.europa.eu/project/id/101097906

or linked to broader knowledge networks. The result is a pattern of digital ephemerality: **valuable research outputs vanish into obscurity**.

To counteract this, ARCHIATER chose to **forgo a custom-built infrastructure**. Instead, **the project embraced Wikidata**—a free, open, multilingual, and structured knowledge base maintained by the Wikimedia Foundation. This decision was not made lightly but built upon previous experiences that revealed both the potentials and the limitations of traditional digital infrastructures.

# Learning from Previous Projects: Toward WikiFAIR

The decision to use Wikidata was shaped by prior experiences in collaborative digital heritage projects. One of these was the digital 3D reconstruction of **Schloss Weikersheim**, part of the initiative "Virtuelle Rekonstruktion: Kulturliegenschaften gestern und heute," a joint effort by the Staatliche Schlösser und Gärten Baden-Württemberg and the Corpus der barocken Deckenmalerei at LMU München.[2] This project used archival documents and architectural analysis to recreate lost historical interiors. While it succeeded in producing compelling scholarly reconstructions, the issue of long-term and open access to the data remained unresolved. After the project's conclusion, the infrastructure in a **WissKI environment** became difficult to maintain.

Another influential project was DeJongeWiki, a digital platform developed to document the architectural history of the **Kasteel van Arenberg** in Heverlee, Belgium.[3] The DeJongeWiki was a side initiative of the Horizon 2020 project PALAMUSTO and the Raymond Lemaire International Centre for Conservation at KU Leuven.[4] It aimed to collate building records, historical images, and scholarly commentary into a comprehensive resource. As with the Weikersheim project, however, questions of long-term sustainability in **a merged Wikibase and MediaWiki environment** loomed large.

These experiences informed the development of **WikiFAIR**[5], a grassroots initiative that advocates for academic data to be hosted within Wikimedia platforms. WikiFAIR emphasizes the value of Wikidata, Wikimedia Commons, and related projects as environments that fulfil the FAIR data principles. By

---

[2] Virtuelle Rekonstruktion: Kulturliegenschaften gestern und heute, https://www.vr-ssg.hs-mainz.de/

[3] DeJongeWiki, mainpage, https://set.kuleuven.be/rlicc/dejongewiki/w/index.php/Main_Page

[4] Webpage of the PALAMUSTO project: https://www.palamusto.eu/blog; webpage of the Raymond Lemaire International Centre for Conservation: https://set.kuleuven.be/rlicc

[5] WikiFAIR, project page on Meta-Wiki, https://meta.wikimedia.org/wiki/WikiFAIR

aligning with open-access infrastructure maintained by a global community, **scholars can ensure that their data lives on**.

ARCHIATER embraced this logic. Rather than investing in another standalone database, the project opted to contribute to the global knowledge graph of **Wikidata—an ecosystem that is both robust and sustainable**.

# Why Wikidata?

Wikidata offers a unique set of features that make it highly suitable as a Virtual Research Environment (VRE). Its **openness**, with all content available under the CC0 license, ensures free and unrestricted access to data, encouraging widespread use and integration. This openness fosters transparency and collaboration across the global research community.

**Sustainability** is another key advantage. Hosted by the Wikimedia Foundation, Wikidata benefits from long-term support, making it more reliable than many other digital platforms that rely on project-specific funding. Researchers can be confident that their data will remain accessible for the long term, addressing a significant challenge in digital humanities—data preservation.

Wikidata also excels in **version control**, as every edit is tracked, allowing scholars to monitor changes and maintain data integrity over time. This feature supports academic rigor by providing a clear record of revisions and facilitating collaborative work.

Its **interoperability** as part of the linked open data (LOD) cloud allows seamless connections with other datasets, enabling data sharing and integration. For interdisciplinary projects like ARCHIATER, this feature is invaluable for connecting research across different fields and regions.

Additionally, Wikidata's **multilingualism** enables researchers from diverse linguistic backgrounds to collaborate effectively. Each entity can have labels and descriptions in multiple languages, promoting international cooperation.

Finally, **community curation** ensures that Wikidata's data remains up-to-date and of high quality. A global network of volunteers and researchers contributes to the continuous improvement and development of entries, enhancing the platform's reliability and accuracy.

For projects like ARCHIATER, which involve collaboration between scholars from different countries and disciplines, these features make Wikidata not just a data repository but **a dynamic platform for collaborative knowledge production**.

# The WikiProject Spital: Structure and Goals

To effectively coordinate its contributions to Wikidata, ARCHIATER launched the **WikiProject Spital**.[6] This specialized initiative was set up to unite scholars who are researching historical hospitals and provides a structured framework for entering and modelling data in a consistent and systematic manner.

One of the primary goals of the WikiProject Spital is to **develop a typology of hospital institutions**, which captures the diversity of these institutions across different periods and geographical locations. This typology allows for a more nuanced understanding of the various types of hospitals that existed in premodern Europe and highlights the differences and similarities in their structures, functions, and roles within society.

Another key objective is to **standardize metadata entries** for each hospital institution. This includes documenting essential information such as the founding year, religious affiliation, architectural features, and the artworks associated with these institutions. By creating uniform metadata, the project ensures that data can be easily compared and integrated, facilitating cross-disciplinary research.

The project also focuses on **building a repository of SPARQL queries**, which will support data analysis and visualization. These queries allow researchers to extract specific data from the Wikidata platform, enabling them to generate visual representations and identify patterns and trends within the historical hospital data. These visualizations help to uncover insights that would otherwise be difficult to see in raw data form.

Finally, the WikiProject Spital aims to **facilitate methodological dialogue** among researchers from various academic traditions. The project provides a platform where scholars with different approaches to studying historical hospitals—whether in the fields of art history, architecture, religious studies, or digital humanities—can collaborate and share their findings. This dialogue encourages a more holistic understanding of hospitals and their role in early modern European societies.

By creating this shared space, WikiProject Spital fosters a collaborative environment where individual insights contribute to a growing corpus of structured, linked data. The project empowers scholars to **contribute locally while thinking globally**, ensuring that their work is part of a broader, interconnected research network. This collaborative approach not only

---

[6] Webpage: https://www.wikidata.org/wiki/Wikidata:WikiProject_Spital

enhances the quality of the data but also encourages ongoing academic exchange and innovation.

## Modeling Hospital Subtypes

One of the central tasks of the project has been to classify and define various hospital subtypes within Wikidata. Premodern hospitals varied significantly in terms of their purpose and structure, reflecting the diverse needs of society across different regions and time periods. To capture this diversity, the project has identified several key hospital subtypes, each with its own distinct characteristics, as examples:

- **Poorhouse (Q1929784):** These institutions were designed to support and provide housing for the dependent and needy. Poorhouses served as a form of social welfare, offering basic care and shelter for those unable to support themselves.

- **Orphanage (Q160645):** Orphanages were residential institutions dedicated to the care of orphans. These facilities were primarily concerned with providing a home and nurturing environment for children who had lost their parents. This term is still used for modern orphanages. Therefore, this category can be used only for entities within historical contexts.

- **Leper colony (Q1406569):** Leper colonies were institutions dedicated to isolating and caring for individuals afflicted with leprosy. These facilities were a response to the widespread fear and stigma surrounding the disease, and their primary function was to segregate those infected to prevent the spread of the illness.

- **Lazaretto (Q859247):** Lazarettos were quarantine facilities established in response to epidemics, especially the plague. These institutions were designed to isolate individuals who had been exposed to contagious diseases, ensuring that the disease did not spread to the broader population.

- **Bimarestan (Q1293141):** The term "Bimarestan" refers to hospitals in the medieval Islamic world. These institutions were often advanced in terms of medical practices and served both as care facilities and centers of medical knowledge.

Each of these hospital subtypes is represented in Wikidata with clearly defined properties and relationships, allowing for a detailed and nuanced

understanding of each type's unique role in society. By modelling these differences explicitly, the project enables richer analysis and meaningful comparisons across time and space, fostering a deeper understanding of how medical care was organized and provided in various historical contexts. This classification also contributes to a more structured approach to historical research, where each institution can be examined within its specific context and compared to similar institutions in different regions or periods.

# Visualizing Data with SPARQL and GIS

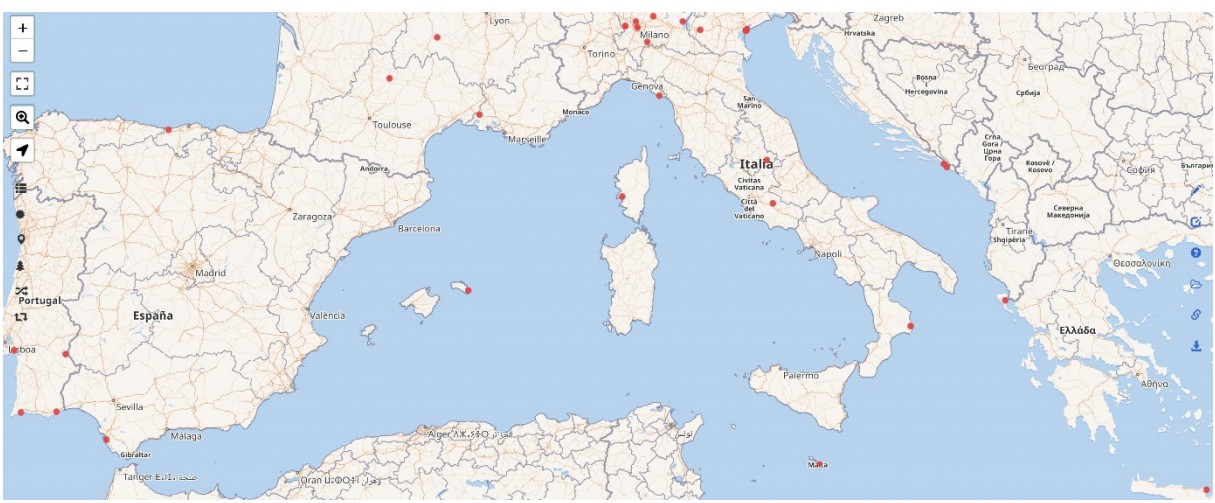

*Figure 2: Distribution of the "Lazzaretto" and "Leprosarium" datasets around the Mediterranean, Interactive GIS map as a visualization of the SPARQL query*

Wikidata's support for SPARQL enables ARCHIATER to conduct quantitative analysis on its dataset. Through custom queries, researchers can generate maps and timelines that reveal historical patterns. For instance:

- Mapping the spread of lazarettos in response to the Black Death and connecting them with researched locations of leper colonies. This approach provides insight into how **the sea** was not only a conduit for trade and communication between societies, but also served as **a barrier to separate and isolate the ill**.[7]

- Visualizing the density of documented **Holy Spirit Hospitals (Q1594896)** in Southern Germany compared to Northern Italy.[8]

- A search for the Late Antique and Byzantine type of **xenodocheion (Q1517445)** suggests that these may no longer be locatable in the eastern

---

[7] Link to SPARQL query: https://w.wiki/DnUm
[8] Link to SPARQL query: https://w.wiki/DrwA

Mediterranean region. Instead, pilgrim accommodations along the Way of St. James in Spain are categorized as such.[9]

These visualizations, often presented as GIS-based maps, allow complex patterns to emerge from what would otherwise remain isolated data points. They are not only valuable for academic research but also serve as powerful tools for teaching and public outreach.

# Modeling Artistic and Social Networks

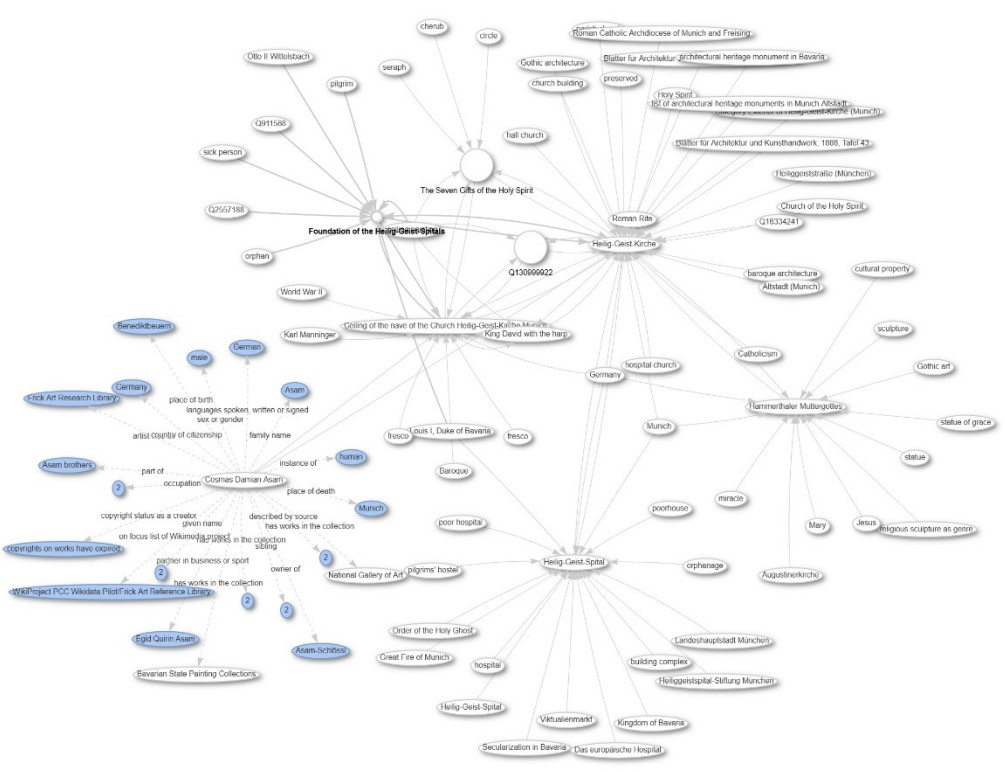

*Figure 3: Relationship network of the painter Cosmas Damian Asam in connection with the Heilig-Geist-Spital in Munich, presented as an interactive graph generated through a SPARQL query*

Quantitative methods in ARCHIATER are significantly enriched by qualitative data modeling, particularly when it comes to representing complex cultural artifacts such as artworks. Wikidata provides the infrastructure to build detailed knowledge graphs that link hospitals to artworks, their patrons, historical events, and associated figures. This is especially valuable in documenting the often-fragmented **histories of artworks** that have undergone changes in location, context, or condition.

Take the example of the **Heilig-Geist-Spital in Munich (Q1594901)**. Using Wikidata, this institution can be linked to the ceiling painting by Cosmas Damian Asam (Q715693), completed in 1727, which was later destroyed during World

---

[9] Link to SPARQL query: https://w.wiki/DrwR

War II. A subsequent reconstruction of the artwork was carried out by Karl Manninger (Q1365373) in 1771. Additional links include the religious order responsible for managing the hospital and its artistic program.[10]

Such modeling is not without challenges. Artworks are not static entities—they may be relocated, altered, restored, or recontextualized in different institutional or liturgical settings. Their physical condition may change over time, and documentation is often incomplete or dispersed. Wikidata allows these multiple, sometimes conflicting, layers of information to coexist through qualifiers, temporal markers, and references. Each element—be it an artist, a patron, a specific date, or a war-related destruction—becomes a node in **a network of semantic relationships**.

This approach allows researchers to trace the life cycle of an artwork across time and space, linking it not only to the institution that originally housed it but also to broader historical and social developments. The resulting knowledge graph enables a level of complexity and nuance that traditional textual scholarship alone often cannot achieve, making Wikidata **a powerful tool for humanities research.**

# International and Interdisciplinary Collaboration

ARCHIATER is fundamentally collaborative, involving researchers in Germany and Italy from diverse disciplinary backgrounds. Wikidata provides a shared platform that bridges methodological and linguistic divides.

Italian researchers contribute detailed archival knowledge and expertise in iconography. German partners bring skills in digital modeling and architectural analysis. The multilingual structure of Wikidata enables each team to work in their preferred language while contributing to a shared data ecosystem. This collaboration also fosters epistemic diversity. Rather than imposing a single interpretive framework, Wikidata allows for multiple, coexisting perspectives.

# Wikidata as Humanities Infrastructure

More than a repository, Wikidata functions as infrastructure for 21st-century humanities research. It provides the foundations for:

- **Sustainable publishing** of research data.
- **Interdisciplinary collaboration** across geographic and institutional borders.

---

[10] Link to SPARQL query: https://w.wiki/DnUs

- **Pedagogical innovation** in digital literacy and data modeling.
- **Public engagement** through integration with Wikipedia and Wikimedia Commons.

By embedding our work in Wikidata, we ensure that it contributes to a growing body of open knowledge. We move beyond the limitations of project-specific platforms to embrace an ecosystem that is truly open, flexible, and future-oriented.

# Conclusion

ARCHIATER demonstrates how **Wikidata can function as a Virtual Research Environment** that supports the full lifecycle of scholarly data—from collection and modeling to analysis and dissemination. It builds upon past lessons and contributes to the future of open, collaborative scholarship.

Rather than creating isolated databases prone to obsolescence, we have chosen to embed our research within an existing, community-maintained infrastructure. This decision reflects a broader commitment to **sustainability, transparency, and epistemic pluralism**.

We call on scholars in the humanities and beyond to consider Wikidata not merely as a repository but as an active partner in the creation and curation of knowledge. In doing so, we move closer to realizing the **principles of FAIR data** —and to building a truly open and interconnected research landscape.

**For more information:**

- WikiProject Spital: https://www.wikidata.org/wiki/Wikidata:WikiProject_Spital
- ARCHIATER Project: https://cordis.europa.eu/project/id/101097906
- WikiFAIR: https://meta.wikimedia.org/wiki/WikiFAIR