# OpenReview forum: "Modeling Architectural Heritage in Wikidata: A Case Study of Early Modern European Hospitals"
_wikimedia.it/Wikidata_and_Research/2025/Conference — WD&R LT_

### Official Review · ~Iolanda_Pensa1 · 2025-01-12
**Interesting topic but not sufficient elements to understand the relevance**

**Originality:** 3
**Impact:** 4
**Confidence:** 3

**Review:**

Exploring the uses and histories of hospitals seems to me very interesting, and I find it relevant and useful to create a link between Wikidata and the research project ARCHIATER - Heritage of Disease: The Art and Architectures of Early Modern Hospitals in European Cities (2024-2028).

The abstract, though, doesn't make any reference to what is already available on Wikidata and what is the new modelling framework for architectural heritage. In Fig 1 I see on a map data about some European hospitals; when I select the items, they do not present data related to building uses, modifications, and art historical objects associated with these institutions. Looking online I found https://archiater.hypotheses.org/1015 and https://www.wikidata.org/wiki/Wikidata:WikiProject_Spital

The abstract doesn't provide sufficient insight to understand what the new framework is about and the case study appears – from the abstract and the info available online – to be in a very early stage, without uploads. There seems to be no example available of this new framework applied to Wikidata  and the new framework is not described on https://www.wikidata.org/wiki/Wikidata:WikiProject_Spital.

WikiFAIR is a nice initiative created by Max Kristen in 2023 and submitted by Max Kristen and Frieder Leipoldto the Wikimedia Foundation https://openreview.net/pdf?id=LFIfMNbz77. it would nice to connect it better with the current project about Open Science and the Wikimedia projects (Daniel Mietchen has worked and is working extensively on this).

**Compliance:**

4

**Notes:**

Maybe the authors can provide an example of Wikidata item updated with the new framework or information related to the framework on https://www.wikidata.org/wiki/Wikidata:WikiProject_Spital. The topic – at the moment not completely developed – could be adequate for a lightening talk or a poster.

**Scientific Quality:**

3

---

### Official Review · ~Alessandra_Boccone1 · 2025-01-21
**Nuovo framework di modellazione per il patrimonio architettonico in Wikidata in fase iniziale**

**Originality:** 4
**Impact:** 4
**Confidence:** 4

**Review:**

Il progetto di modellazione di un nuovo framework in Wikidata per il patrimonio architettonico relativo agli ospedali (considerati punti di riferimento architettonici e sociali) è sviluppato all'interno della cornice dell'ERC ARCHIATER - Heritage of Disease: The Art and Architectures of Early Modern Hospitals in European Cities: il legame e il confronto con il contesto accademico è sicuramente un punto a favore della proposta, che dimostra un potenziale interessante, ma che sembra essere ancora in una fase embrionale. Sarebbe interessante capire se il lavoro è andato avanti e come si è sviluppato in concreto, per poter dare un giudizio più completo. Oltre all'arricchimento degli item, colpisce il fatto che vengano utilizzati anche i tool di visualizzazione e analisi dei dati, cosa che sottintende una buona conoscenza non solo di Wikidata, ma anche di tutta una serie di strumenti più complessi a corredo del database.

**Compliance:**

4

**Scientific Quality:**

3

---

### Official Review · ~Camillo_Carlo_Pellizzari_di_San_Girolamo1 · 2025-01-22
**A good workflow for data curation in Wikidata**

**Originality:** 4
**Impact:** 3
**Confidence:** 4

**Review:**

The paper introduces a new data model for early modern European hospitals in Wikidata, analyses the challenges in improving the data already added to Wikidata and explores how Wikidata could allow to create data visualizations and analyses from the data: this workflow is promising and could be applied also to other fields of research.
However, since presently the data model has been applied only to a small number of items in Wikidata (as far as https://www.wikidata.org/wiki/Wikidata:WikiProject_Spital shows), its future impact on Wikidata is not fully clear. The project seems in an early phase.

**Compliance:**

5

**Notes:**

The proposal could be transformed into a lightning talk (or a poster).

**Scientific Quality:**

4

---

### Decision · Program_Chairs · 2025-02-05

**Decision:**

Accept (LT)

**Comment:**

==New format: LT==

Dear Authors,
thank you very much for your proposal. We regret to inform you that your proposal was not selected among the papers.

Even if not selected as paper, we consider your proposal relevant and interesting and we would like to propose you to prepare instead a lightening talk (if you - or another member of your team - can participate in presence at the conference) or a poster (which can be exhibited even if you will not attend the conference).

It would be a pleasure to learn more about your work through a lightening talk or a poster.
Thank you for submitting a proposal and please let us know if you like the idea of converting it into a lightening talk or a poster and which format you prefer.

Regards,
The scientific committee of the conference Wikidata and Research